# Inhibition of Diarrheal Shellfish Toxins Accumulation in the Mussel *Perna viridis* by Curcumin and Underlying Mechanisms

**DOI:** 10.3390/toxins13080578

**Published:** 2021-08-20

**Authors:** Kuan-Kuan Yuan, Guo-Fang Duan, Qing-Yuan Liu, Hong-Ye Li, Wei-Dong Yang

**Affiliations:** Key Laboratory of Aquatic Eutrophication and Control of Harmful Algal Blooms of Guangdong Higher Education Institute, College of Life Science and Technology, Jinan University, Guangzhou 510632, China; yk9723@stu2019.jnu.edu.cn (K.-K.Y.); dgf1633071004@stu2016.jnu.edu.cn (G.-F.D.); lqy178788@stu2019.jnu.edu.cn (Q.-Y.L.); thyli@jnu.edu.cn (H.-Y.L.)

**Keywords:** DSTs, *Perna viridis*, AhR, HR96, CYP3A4, CUR

## Abstract

Diarrheal shellfish toxins (DSTs) are among the most widely distributed phytotoxins, and are associated with diarrheal shellfish poisoning (DSP) events in human beings all over the world. Therefore, it is urgent and necessary to identify an effective method for toxin removal in bivalves. In this paper, we found that curcumin (CUR), a phytopolylphenol pigment, can inhibit the accumulation of DSTs (okadaic acid-eq) in the digestive gland of *Perna viridis* after *Prorocentrum lima* exposure. qPCR results demonstrated that CUR inhibited the induction of DSTs on the aryl hydrocarbon receptor (AhR), hormone receptor 96 (HR96) and CYP3A4 mRNA, indicating that the CUR-induced reduction in DSTs may be correlated with the inhibition of transcriptional induction of AhR, HR96 and CYP3A4. The histological examination showed that *P. lima* cells caused severe damage to the digestive gland of *P. viridis*, and the addition of curcumin effectively alleviated the damage induced by *P. lima*. In conclusion, our findings provide a potential method for the effective removal of toxins from DST-contaminated shellfish.

## 1. Introduction

Phytotoxin accumulation in filter-feeding shellfish is a global phenomenon. Bivalve mollusks are undoubtedly the main carriers of shellfish toxins, including diarrheal shellfish toxins (DSTs) and paralytic shellfish toxins (PSTs) [1]. Severe poisoning may occur when people consume shellfish that are contaminated with such phytotoxins [2]. DSTs are among the most frequent and globally distributed marine biotoxins, and are usually associated with diarrheal shellfish poisoning (DSP) events all over the world [3,4,5]. They include okadaic acid (OA) and dinophysis toxins (DTXs). Some genera of *Prorocentrum* and *Dynophysis* can produce DSTs [3]. Among them, *Prorocentrum lima* has been widely used in DST-related studies as a reliable source of DSTs [6].

The main route of human exposure to DSTs is through filter-feeding shellfish. The consumption of these contaminated shellfish may cause food poisoning and obvious gastrointestinal dysfunction, such as diarrhea, abdominal cramps, nausea, vomiting and other symptoms [7,8]. Marine filter feeders such as mussels, oysters, clams, and scallops can accumulate these biotoxins in their tissues, even in environments with a low concentration of DST-producing algal species [9]. More importantly, the properties of these toxins do not change with cooking or freezing, nor do they affect the taste of the contaminated shellfish; thus, they are often difficult to detect without rigorous testing [10,11]. How to effectively reduce the accumulation of toxins in shellfish has attracted widespread attention, and various methods for reducing the accumulation of toxins in shellfish have been suggested [12,13]. Nevertheless, the safety and feasibility of these methods in practical application are open to question.

Several studies have demonstrated that cytochrome P450 (CYP) may be involved in the metabolism of DSTs [14,15]. Chi et al. observed the OA-caused significant up-regulation of CYP3A4 mRNA in the bay scallop *Argopecten irradians* [15]. Previously, we found that ketoconazole, an inhibitor of CYP3A4, can decrease OA content in the digestive gland of the mussel *Perna viridis* [16]. However, ketoconazole may cause certain side effects in humans, limiting its practical use in reducing toxin accumulation [17]. Subsequently, we found that cinnamaldehyde, a compound present in several edible substances, reduced the accumulation of DSTs in mussels, and interestingly, CYP3A4 expression was also significantly down-regulated by cinnamaldehyde in *P. lima*-exposed mussels [18]. These results suggest that CYP3A4 may be an important participant in the metabolism of DSTs, although the related mechanism remains unclear. 

CYP3A4 is transcriptionally regulated by several nuclear receptors, such as the aryl hydrocarbon receptor (AhR), pregnane X receptor (PXR) and constitutive androstane receptor (CAR) [19]. Numerous studies have shown that AhR is a key regulator of the metabolism of exogenous chemicals in multiple species [20]. Many chemicals can function as ligands to form complexes with AhR, thereby regulating expression of target genes such as CYP1 and many others that are primarily involved in the detoxification and transport of xenobiotics and drugs [21,22]. Zanette et al. found that AhR activators, naphthoflavones and polychlorinated biphenyls induced the expression of CYP3-related genes in the mussel *Mytilus edulis* [23]. Nuclear receptors, such as PXR, CAR, and vitamin D (1, 25-dihydroxyvitamin D3) receptor (VDR), can bind to RXR to form a heterodimer, thereby regulating the transcription of CYP3A genes [24]. Hormone receptor 96 (HR96), a presumptive toxicant receptor in the NR1J group of invertebrate nuclear receptors, is evolutionarily closely related to the NR1I (VDR/CAR/PXR) group in mammals [25]. A growing body of research has shown that NR1I group can regulate the expression of phase I, II and III detoxification genes during the metabolism of xenobiotic substances [26,27,28]. 

Curcumin (CUR) is a phytopolylphenol pigment isolated from the plant *Curcuma longa*. As a natural pigment with high safety recognized by Food and Agriculture Organization of the United Nations (FAO) and World Health Organization (WHO), CUR has a variety of anti-inflammatory and anti-cancer properties [29,30]. Interestingly, high concentrations of curcumin almost completely inhibited the activity of CYP3A4 in vitro [31]. Some studies have demonstrated that CUR can increase the bioavailability of tamoxifen, probably due to the inhibition of CYP3A4-mediated metabolism of this compound in the small intestine [32]. CUR may also affect the expression of CYP3A4 in shellfish, thereby reducing the toxin accumulation.

To explore whether CUR is able to reduce DST accumulation in bivalve mollusks and to reveal the underlying mechanism, we analyzed the effect of CUR on the accumulation of DSTs in the mussel *P. viridis* after *P. lima* exposure under laboratory conditions. The expression, and histological changes in the digestive gland of mussel were observed.

## 2. Results

### 2.1. CUR Decreases the Accumulation of DSTs in the Digestive Gland but Not Gills of Mussels

As shown in Figure 1, the addition of CUR had a great impact on the accumulation of DSTs in the digestive gland of mussel after exposure to *P. lima*. When the concentrations of CUR were 10 μmol L^−1^ and 20 μmol L^−1^, the DST levels in the digestive gland were significantly lower than that in the control (*p* < 0.05).

Figure 2 demonstrates the DST levels in the digestive gland and in the gills of the *P. lima*-exposed mussels at different times after the addition of CUR (20 μmol L^−1^). The DST content in the *P. lima*-exposed mussels was higher than the control counterparts at both 12 and 48 h (*p* < 0.05). In the digestive gland, the content of DSTs in the *P. lima*-exposed mussels with CUR was significantly lower than in those without CUR at 12 and 48 h (*p* < 0.05). In contrast, there was no significant difference in the DST content in the gills between the *P. lima*-exposed mussels with and without CUR.

To further explore the fate of DSTs in the digestive gland, we measured the level of DSTs in culture medium. As expected, the DST content in the digestive gland was decreased in the *P. lima* + CUR group at 12 h and 48 h, while the DST content in culture medium increased compared with the treatment without CUR.

### 2.2. CUR Reduced Damage to the Digestive Gland in the P. lima-Exposed Mussels

Figure 3 shows the histological alterations in the digestive gland after exposure to *P. lima* with CUR and without CUR. In the control group, the columnar epithelial cells of the digestive diverticulum were neatly arranged, the digestive tubes were interconnected by connective tissue, and there was no atrophy of the epithelial cells. Meanwhile, no significant structural changes were observed in the CUR group. In contrast, the digestive gland tissues in the *P. lima*-exposed mussels showed significant structural changes (Figure 3C). 

At 12 h, the epithelial cells were severely atrophied, some digestive tubules disappeared, and the digestive diverticula formed vacuoles. At 48 h, the epithelium cells displayed more severe atrophy, with partial decomposition of the epithelium, disappearance of the digestive cells, and diverticula deformation of the digestive gland. However, in the *P. lima*-exposed mussels with curcumin, only mild atrophy of the epithelial cells was observed, and no epithelial cell decomposition was observed. These findings suggest that CUR may effectively alleviate the damage caused by DSTs to the digestive gland in mussels, which might provide new evidence that CUR reduced the accumulation of DSTs in the digestive gland in mussels.

### 2.3. CUR Might Affect the Metabolic Process of DSTs in Mussels by Altering the Expression of Genes Associated with Metabolism

As illustrated in Figure 4, the expression of CYP3A4 in the *P. lima*-exposed mussels (*P. lima* group) was increased compared to the control group (*p* < 0.05) at 48 h. Meanwhile, the expression of AhR at 12 h, and HR96 at 48 h were significantly up-regulated in the *P. lima*-exposed mussels compared to the control group (*p* < 0.05). However, after the addition of CUR, CYP3A4 transcription was significantly decreased compared with the counterpart without CUR (*P. lima* group) at 12 and 48 h. In addition, the expressions of AhR at 12 h, and HR96 at 48 h after the addition of CUR (*P. lima* + CUR group) were remarkably lower than those without CUR (*P. lima* group). These results suggest that CUR may affect the metabolic process of DSTs in mussels by altering the expression of genes involved in metabolism, such as AhR, HR96 and CYP3A4.

## 3. Discussion

At present, a variety of methods to remove shellfish toxins have been proposed, including physical, biological and chemical methods [11,33]. However, these methods are difficult to implement in practice and/or their safety is questionable. Previously, we found that ketoconazole inhibited the expression of CYP3A4, and reduced the OA content in the digestive glands in the *P. lima*-exposed mussel [16]. Subsequently, we examined the possibility that edible natural substances would reduce the accumulation of DSTs in mussels and found that cinnamaldehyde decreased the accumulation of DSTs in the digestive glands of mussels, and the CYP3A4 transcript was significantly down-regulated [18]. The accumulation of DSTs may be related to the expression of CYP3A4, though the underlying mechanism remains unclear.

In the current study, we found that CUR significantly suppressed the accumulation of DSTs in digestive glands of the *P. lima*-exposed mussels. The histological alteration further demonstrated that CUR effectively alleviated the *P. lima*-induced damage to the digestive gland. To elucidate the fate of reduced DSTs in the digestive gland after the addition of CUR, we detected the DST content in gills and in culture medium (seawater). In line with our expectation, the DST content in culture medium was increased significantly in the *P. lima*-exposed group with CUR, while the content of DSTs in gills was not significantly changed. These outcomes suggest that CUR may accelerate the elimination of DSTs rather than transfer them to other tissues.

Histomorphology provides a means to visualize the impact of toxin accumulation on the health of shellfish [34]. Several studies have dealt with the damage caused by DSTs and DST-producing dinoflagellates to bivalves [35,36]. The main lesions in bivalves were tubular atrophy of digestive gland diverticula, the infiltration of hemocyte, the reduction in digestive cells, and the atrophy and decomposition of epithelial cells [35,36]. Consistent with the previous observations, the digestive diverticula in the digestive gland of the *P. lima*-exposed mussel showed tubular atrophy, and the loss of digestive cells resulted in a substantial reduction in epithelial cells. The digestive cells have a well-developed endo-lysosomal system for the digestion of xenobiotic substances [37,38]. Recovery from epithelium atrophy caused by loss of digestive cells is slow or even impossible under natural conditions [39]. However, the addition of CUR effectively prevented the damage caused by DSTs to the digestive gland, which provided new evidence that CUR reduced the accumulation of DSTs in the digestive gland in mussels.

Detoxification in bivalve mollusks is a very complicated process, and cytochrome P450 is involved in the oxygenation of xenobiotic metabolism [40]. Ferron et al. have shown that CYP3A4 mediates the metabolism of lipophilic algal toxins, thereby reducing their in vitro toxicity [41]. The CYP3A proteins in bivalve tissues such as the digestive gland and gills have been demonstrated [42]. Furthermore, some studies have proposed that CYP3A4 might contribute to the metabolism of DSTs in mussels [15,16,18]. Interestingly, based on the previous findings [19,23,43], CYP3A4 in bivalves may be transcriptionally regulated by several nuclear receptors such as AhR and HR96. So, we observed the changes in expression of CYP3A4, CYP3A1, AhR and HR96.

We found that the expression of CYP3A4 in the digestive gland was increased at 48 h. Meanwhile, the expression of AhR at 12 h and HR96 at 48 h were significantly up-regulated. However, CYP3A4 transcription in the *P. lima*-exposed mussel with CUR (*P. lima* + CUR group) was significantly lower than that without CUR (*P. lima* group) at both 12 and 48 h, indicating that CUR reduced the expression of CYP3A4 in the digestive gland induced by DSTs. Accordingly, the expressions of AhR and HR96 were significantly decreased in the *P. lima*-exposed mussels with CUR, and were lower than in mussels without CUR. It is likely that CUR down-regulates the expression of CYP3A4 by suppressing the expression of AhR and HR96 induced by DSTs [44,45,46]. During the metabolism of DSTs, the expression of AhR was only induced at 12 h and HR96 was induced at 48 h. The expression of AhR and HR96 seemed to exhibit temporal discontinuity, suggesting that AhR and HR96 may be alternately activated [47]. So far only a few studies have been performed on nuclear receptors in shellfish, and very limited information is available on the regulatory effects of nuclear receptors on genes related to detoxification [48,49]. Recent studies have demonstrated that there are great differences between detoxification gene modules in different species [50], and the potential regulatory mechanism needs to be verified in the future.

In this study, curcumin was found to be effective in reducing the accumulation of DSTs in the digestive glands of the *P. lima*-exposed mussels and in mitigating the damage caused by DSTs to mussels. However, whether it can effectively reduce the accumulation of toxins in actual seawater needs further study. In addition, changes in the expression of CYP3A1 in the presence of curcumin suggests the complexity of curcumin in modulating the metabolism of xenobiotic compounds in shellfish. Therefore, further studies, especially on the removal mechanism, ecological safety and economic cost, should be conducted before attempting to use curcumin to remove toxins in shellfish in field. In addition, the fate of the DSTs from bivalves is indeed an interesting question that deserves further study.

## 4. Conclusions

Curcumin could significantly reduce the accumulation of DSTs in the digestive gland of mussels after *P. lima* exposure. After exposure to *P. lima*, CYP3A4, AhR and HR96 were expressed in the digestive gland of mussels, while CUR inhibited the overexpression. It is likely that CUR could down-regulate the expression of CYP3A4 by suppressing the expression of AhR and HR96. These may lead to changes in the metabolic process of DSTs in vivo, and ultimately reduce the accumulation of toxins in the digestive glands of *P. lima*-exposed mussels; thus, effectively ameliorating the damage to the digestive gland. However, the removal mechanism remains largely unclear, especially the exact role of CYP3A4, which needs further study.

## 5. Materials and Methods

### 5.1. Materials and Animals

The mussel *P. viridis* used in the experiment was purchased from the Huangsha Aquatic Products Market in Guangzhou, and the mussels were collected from the Zhanjiang Sea area, China. *P. viridis* individuals with a length of 8 to 10 cm and a total soft body weight of 12.4 ± 1 g were chosen and transported to our laboratory as quickly as possible under low temperature. After the mud on the shell surface was removed, the cleaned animals were cultured in several aerated glass tanks (45 cm × 30 cm × 35 cm) under controlled conditions (17.5 ± 1 °C, 12 h light/12 h dark cycle). The natural seawater with a salinity of 35 ± 2, was changed regularly every day, and the chlorophyte *Tetraselmis subcordiformis* (1 × 10^7^ cells/L) was used to feed the mussels. Animals with good growth and strong foot silk were selected for the experiments.

The chlorophyte *T. subcordiformis* used in the experiment was kindly provided by the Institute of Hydrobiology, Chinese Academy of Sciences. As in previous studies, the DST-producing dinoflagellate *P. lima* (CCMP 2579), which has been proved to produce OA and DTX1 [51,52], was purchased from the National Center for Marine Algae and Microbiology (NCMA) (https://ncma.bigelow.org, Accessed on 17 August 2021). The two algal species were cultured in f/2 silicon-free medium, which was filtered through a 0.22 µm filter. The cultures were grown at 20 ± 1 °C in an artificial climate incubator (Ningbo Jiangnan Instrument Factory, China) (60 µmol m^−2^ s^−1^, 12 h/12 h photoperiod).

Curcumin (purity ≥ 94%) purchased from Sigma-Aldrich (C7727, Germany), was dissolved in 0.5 mol L^−1^ NaOH solution to a concentration of 200 mmol L^−1^, and then diluted to 80 mmol L^−1^ with natural seawater [53]. 

### 5.2. Experimental Design

Given that DSTs are mainly concentrated in the digestive gland of mussels [54], we focused on the DSTs content in digestive gland tissue of the *P. lima*-exposed *P. viridis*. Our experiment was divided into two steps:

Step I: Changes in DSTs content (OA-eq) after the addition of CUR. The *P. viridis* individuals were fed with 1 × 10^7^ cells/L of *T. subcordiformis* and 2 × 10^6^ cells/L of *P. lima* for 2 h, CUR was added to the culture medium (6 L) with different final concentrations (0, 1, 5, 10, 20, 50 µmol L^−1^, respectively). All digestive gland tissues with a weight of about 0.1 g were collected from each animal at 48 h after the addition of CUR or solvent. To reduce the individual differences, tissues from six individual mussels within the same group were pooled together as one sample for the detection of DSTs. Therefore, each treatment group had three samples, each of which contained six individual mussels. 

Step II: Alterations in the DST content in mussels after exposure to *P. lima* for different times with or without CUR. The mussel *P. viridis* individuals were randomly divided into four groups as follows: (1) Control, *T. subcordiformis* (1 × 10^7^ cells/L); (2) CUR, *T. subcordiformis* (1 × 10^7^ cells/L) *+* CUR (20 μmol L^−1^); (3) *P. lima*, *T. subcordiformis* (1 × 10^7^ cells/L) + *P. lima* (2 × 10^6^ cells/L); (4) *P. lima +* CUR, *T. subcordiformis* (1 × 10^7^ cells/L) + *P. lima* (2 × 10^6^ cells/L) *+* CUR (20 μmol L^−1^). In terms of design, the final concentration of 20 µmol L^−1^ CUR was added into culture medium at 2 h after 1 × 10^7^ cells/L of *T. subcordiformis* and 2 × 10^6^ cells/L of *P. lima* were added. The control group and *P. lima* group was added with isovolumetric solvent (mixture of 600 µL of 0.5 mol L^−1^ NaOH and 900 µL of seawater). For the histological examination, digestive gland tissues were sampled at 12 and 48 h after the addition of CUR. At 12 h and 48 h after the addition of CUR, eighteen individuals were randomly sampled at each time point in each treatment. Gills and digestive gland with a weight of about 0.1 g/individual were excised, and tissue from six individuals at each time point within the same treatment were pooled together as one sample for DSTs (stored in −20 °C freezer) and total RNA extraction (stored in −80 °C freezer). In addition, culture medium (seawater sample) was collected at 12 h and 48 h (stored in 4 °C freezer). 

### 5.3. Detection of DSTs

DST extraction and detection in shellfish and culture medium was performed as per our previous paper [16]. An okadaic acid (DSP) ELISA test kit (Abraxis, Warminster, PA, USA) was used, in which the absorbance at 450 nm was detected by a Synergy H1 Hybrid Multi-Mode Microplate Reader (Bio-Tek, Winooski, VT, USA). The OA content was calculated according to the standard curve generated and expressed as ng OA eq g^−1^.

The DST content in the culture medium was measured using an Oasis HLB 6 cc (500 mg) LP Extraction Cartridge (Waters Ltd., Worcester, MA, USA) filter column as described by Fang et al. with slight modifications [55], in which a Supelco Visiprep 12 tube anti-contamination (DL) solid phase extraction unit was used. Briefly, the HLB filter column was firstly activated with methanol, and washed with distilled water. Then, 30 mL of culture medium sample was added to the HLB filter column. The column was eluted successively with 3 mL 100% methanol and 2 mL 30% methanol. The collected eluent was dried with nitrogen and dissolved in 500 µL of methanol for DST detection. DST detection was carried out via ELISA as above.

### 5.4. qPCR

Total RNA was extracted from an approximately 50 mg sample of mussel tissue by using a Total RNA Kit I (50) R6934-01 (Omega, Norcross, GA, USA). RNA integrity and genome DNA contamination were tested by a denaturing agarose gel electrophoresis. RNA concentration was measured using a NanoDrop 2000 Spectrophotometer at 260 nm (Implen, Munich, Germany). Approximately 1 µg of total RNA was reverse transcribed using a HiScript^®^ II Q RT SuperMix for qPCR (+gDNA wiper) (R223-01, Vazyme, Nanjing, China) kit according to the manufacturer’s instructions.

The most stable genes, RPL3 and RPL13 were used as reference genes to normalize the expression of target genes including AhR, HR96, CYP3A4 and CYP3A1. The expression of five candidate genes including elongation factor 1 alpha (EF1α), ribosomal protein L3 (RPL3), ribosomal protein L13-like (RPL13), ribosomal protein L37 (RPL37) and ubiquitin A-52 (UBA52) was examined for screening reference genes. Three kinds of software including geNorm [56], NormFinder [57] and BestKepper [58] were employed. All the primers were designed based on the sequences of *Mytilus edulis* by using Primer 5.0 software (Premier, Waterloo, ON, Canada) according to the following parameters: ideal melting temperature (Tm), GC content, length, and amplicon length (Table 1).

PCR was performed on a CFX96 Real-Time PCR System (Bio-Rad, Hercules, CA, USA) using AceQ^®^ qPCR SYBR^®^ Green Master Mix (Vazyme, Nanjing, China) with the following profile: 95 °C for 5 min, 40 cycles of 95 °C for 10 s, 60 °C for 30 s. PCR specificity was evaluated by melting curve analysis from 65 °C to 95 °C, increasing by 0.5 °C every 5 s. The reaction mixture (20 µL) consisted of 10 µL of AceQ^®^ qPCR SYBR^®^ Green Master Mix, 7.6 µL of H_2_O, 0.2 µL of each of the forward and reverse primers (10 µmol L^−1^) and 2 µL of the first-strand cDNA template from 1 µg RNA. Amplification was also performed on equivalent double-distilled water as no template control (NTC) to check the absence of contaminant. The inter-run calibrator (IRC) was set to eliminate errors in running data for the two boards. The amplification efficiency of all genes assayed was guaranteed to be between 98% and 105%, and the correlation coefficient was 0.99.

The comparative C_q_ method was used to analyze relative expression level of genes as described by Derveaux et al., where multiple reference genes and inter-run calibration algorithms were considered [59].

### 5.5. Histological Examination

Histological examination was performed as described by previous reports with some modifications [35]. Briefly, digestive gland tissue dissected from animals was fixed in 4% paraformaldehyde solution for at least 48 h. Subsequently, the tissue was dehydrated in gradient ethanol solutions (from 75% to 95%), then embedded in paraffin. The embedded tissue was sectioned at 4 µm thickness with a manual rotary microtome (Leica RM2235, Lecia Microsystems Nussloch Gmbh, Heidelberger, Germany). The paraffin-embedded section obtained was deparaffinized in xylene, rehydrated in gradient ethanol solutions, and then stained with hematoxylin and eosin. After each sample was sealed with neutral balsam, the stained slice was imaged using a Panoramic MIDI Slide Scanner (3DHISTECH, Budapest, Hungary), and analyzed by Case-Viewer software (3DHISTECH, Budapest, Hungary).

### 5.6. Statistical Analyses

Statistical analyses were conducted by using SPSS Statistics 25.0 for Windows (IBM SPSS Statistics 25.0 Packet program, Arnomk, NY, USA) (https://www.ibm.com/cn-zh, Accessed on 17 August 2021.) All data were expressed as mean ± SD. In Step I, the difference in DST content between two groups was examined by *t*-test. In Step II, differences between treatment groups were estimated by Fisher’s protected multiple comparisons least significant difference (LSD) test with a significant difference at *p* < 0.05.

## Figures and Tables

**Figure 1 toxins-13-00578-f001:**
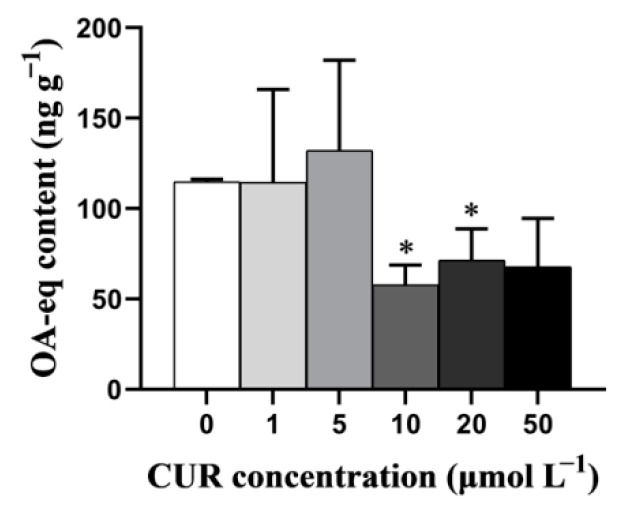
DST content in the digestive gland of the *P. lima*-exposed mussels with or without curcumin. Significant differences compared to control (with 0 μmol L^−1^ CUR) are represented by asterisks (*t*-test, * *p* < 0.05). Data are presented as mean ± SD (*n* = 3).

**Figure 2 toxins-13-00578-f002:**
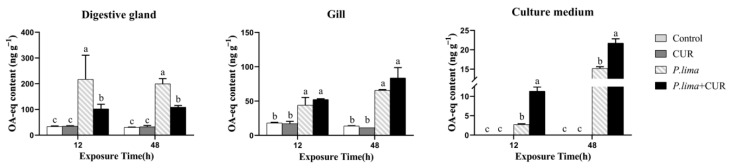
DST content in the digestive gland and gills of the *P. lima*-exposed mussels and in culture medium at different times after the addition of CUR (20 μmol L^−1^). Control, *T. subcordiformis* (1 × 10^7^ cells/L); CUR, *T.*
*subcordiformis* (1 × 10^7^ cells/L) *+* CUR (20 μmol L^−1^); *P. lima*, *T. subcordiformis* (1 × 10^7^ cells/L) + *P. lima* (2 × 10^6^ cells/L); *P. lima +* CUR, *T. subcordiformis* (1 × 10^7^ cells/L) + *P. lima* (2 × 10^6^ cells/L) *+* CUR (20 μmol L^−1^). Data are presented as mean ± SD (*n* = 3). Bars of the respective treatments followed by the same letters indicates that the difference is not significant at *p* < 0.05 (Fisher′s protected multiple comparisons LSD test).

**Figure 3 toxins-13-00578-f003:**
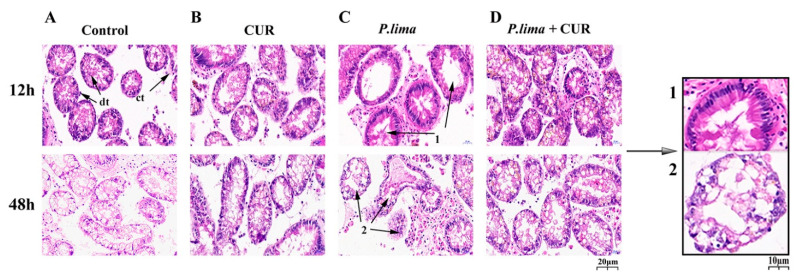
Histological sections of digestive glands of *Perna viridis* at 12 h and 48 h (HE staining, ×400). (**A**) Control, *T. subcordiformis* (1 × 10^7^ cells/L); (**B**) CUR, *T. subcordiformis* (1 × 10^7^ cells/L) + CUR (20 μmol L^−1^); (**C**) *P. lima*, *T. subcordiformis* (1 × 10^7^ cells/L) + *P. lima* (2 × 10^6^ cells/L); (**D**) *P. lima* + CUR, *T. subcordiformis* (1 × 10^7^ cells/L) + *P. lima* (2 × 10^6^ cells/L) + CUR (20 μmol L^−1^); dt, alimentary canal; ct, connective tissue. Marker 1, severe atrophy of epithelial cells; Marker 2, disintegration of epithelial cells, destruction of digestive ducts, and malformation of digestive gland diverticulum.

**Figure 4 toxins-13-00578-f004:**
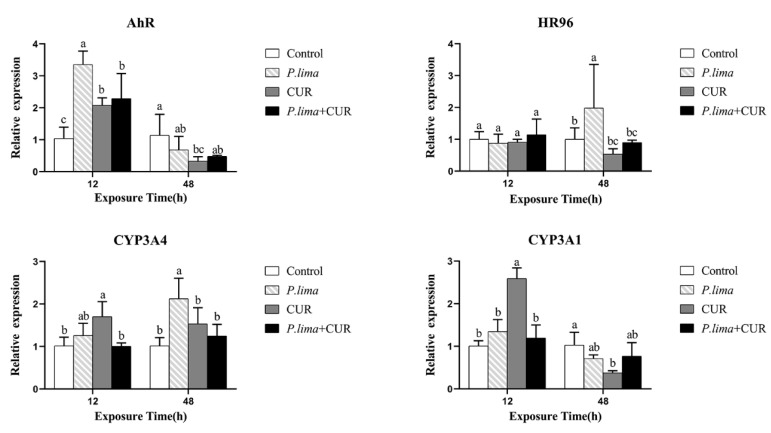
Expression levels of genes associated with the metabolism of xenobiotic compounds in the digestive gland of the *P. lima*-exposed mussels as shown by qPCR. Control, *T. subcordiformis* (1 × 10^7^ cells/L); *P. lima*, *T. subcordiformis* (1 × 10^7^ cells/L) + *P. lima* (2 × 10^6^ cells/L); CUR, *T.*
*subcordiformis* (1 × 10^7^ cells/L) + CUR (20 μmol L^−1^); *P. lima +* CUR, *T. subcordiformis* (1 × 10^7^ cells/L) + *P. lima* (2 × 10^6^ cells/L) *+* CUR (20 μmol L^−1^). Data presented as mean ± SD (*n* = 3). Bars of the respective treatments followed by the same letters indicates that the difference is not significant at *p* < 0.05 (Fisher′s protected multiple comparisons LSD test).

**Table 1 toxins-13-00578-t001:** Primer for qPCR.

Gene Name	Primer Sequence (5′–3′)		Amplicon Size (bp)
EF1-Fα	F:	CACTCCGTCTTCCACTCCA	131
	R:	CCTCTGGCATTGACTCGTG	
Tubulin-β	F:	AGGAAGGAGGCTGAGAGTTGT	135
	R:	TTTGGAGATGAGCAGGGTTC	
RPL3	F:	GGTGGCACTATCTCCCAGAA	98
	R:	GCCATCTGGACGTTACACCT	
UBA52	F:	TTACATTTGGTCCTGCGTCTC	135
	R:	CAGTTGGTAGCCCTTTGATGA	
RPL13	F:	TAAAGACTGGCAACGCTATGT	155
	R:	TCACAACTGGTCGGAGAAG	
RPL37	F:	GTCGCAATAAGACGCACACGTTG	179
	R:	GTGCCTCATTCGACCAGTTCCG	
AhR	F:	GAGAGTTCTGGAGTGTTCGTGGAC	109
	R:	TGAGGACACGGTTCAGTCAG	
HR96	F:	TGGAGCCTGATGAAGAATACC	194
	R:	ACTGAGAGAGAGCCTTGAATG	
CYP3A4	F:	ACCAATGCAAATGCGTGGTC	99
	R:	TCAGCAGGCAGTATTCCAGGAGAG	
CYP3A1	F:	CGCTGCTGTGACGATCTGGTAG	141
	R:	TCTCTGCGAATTCACCTGCAACC	

EF1α, elongation factor 1 alpha; RPL3, ribosomal protein L3; RPL13, ribosomal protein L13-like; RPL37, ribosomal protein L37; UBA52, ubiquitin A-52; CYP3A1, Cytochrome P450 3A1; CYP3A4, Cytochrome P450 3A4. AhR, Aryl hydrocarbon receptor; HR96, Nuclear hormone receptor 96.

## Data Availability

The data in this study are available from the corresponding author upon request.

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
