# Peer review of "Inhibition of Diarrheal Shellfish Toxins Accumulation in the Mussel Perna viridis by Curcumin and Underlying Mechanisms"

_toxins, 2021, doi:10.3390/toxins13080578_

Round 1

Reviewer 1 Report

Review of “Inhibition of diarrheal shellfish toxins accumulation in the mussel Perna viridis by curcumin and underlying”

Overall, there are some things that need to be confirmed about toxicity testing, as the interpretation of the results can be quite different. The scale of the feeding condition for mussel is unclear for the experiments in Figure 1 and Figure 2. The cell densities of non-toxic algae and toxic dinoflagellate are shown (T. subcordiformis (1×107 cells/L) + P. lima (2×106 cells/L)), but the seawater volume of the experimental vessel is not shown. The experiment was conducted by feeding for 2 hours and then adding CUR, but I cannot confirm whether or not all the algae fed within the first two hours were given to the mussels. Is there any possibility that the algae was not sufficiently removed during the 2-hour feeding period because the feeding experimental temperature has been reached within the temperature range where mussel feeding activity appears to be too low. (Perna viridis is a tropical species and therefore the optimal rearing temperature is approximately exceeded 25°C), therefore, water filtering activity may still continue after CUR is added. Furthermore, no basic data are provided on the effect of the presence of 20 μM of CUR on filter water activity in the mussels. Is it possible that unfiltered algae are still present after the addition of CUR and that the addition of CUR has had an inhibitory effect on filter water activity to mussels, resulting in a less P. lima accumulation and low levels of DST detected in the gut? It is also possible that the low accumulation of P. lima caused lower damage to the gut. The volume of rearing water and the average filtrate volume at 17.5 °C of mussel should be shown to prove that all algae had been removed within 2 hours feeding experiments.

Although the concentration of toxins in seawater has been measured, this is for the dissolved form. Since the exposure density of P. lima is much higher than that found in nature, and there is no difference in toxin concentration in the gill, it is possible that the test was conducted under conditions where much of the algae fed was simply discarded as pseudofeces. Generally, the fate of the toxin should be measured both fecal and dissolved forms to distinguish between excreted from the once accumulated toxin and rejected material as peudofeces.

The authors also found damage to the intestine in histological observations, but we do not know whether this was caused by DST. This is because some dinoflagellate species, including genus Prorocentrum, have uncharacterized toxic substances that have long been known to cause intestinal damage when fed, even in dinoflagellate species that do not produce DST (see below). P. lima also has possibilities to have similar uncharacterized toxic substances, and the effects of uncharacterized toxic substances may be more apparent in tissue observations than DST-induced tissue damage.

Gary H. Wikfors and Roxanna M. Smolowitz (1995) Experimental and histological studies of four life-history stages of the Eastern oyster, Crassostrea virginica, exposed to a cultured strain of the dinoflagellate Prorocentrum minimum. Biological Bulletin , 188(3), 313-328

I Pearce, JH Handlinger, GM Hallegraeff (2005) Histopathology in Pacific oyster (Crassostrea gigas) spat caused by the dinoflagellate Prorocentrum rhathymum. Harmful Algae, 4(1), 61-74

Roxanna Smolowitz and Sandra E. Shumway (1997) Possible cytotoxic effects of the dinoflagellate, Gyrodinium aureolum, on juvenile bivalve molluscs. Aquaculture International, 5, 291–300

Round 2

Reviewer 1 Report

Revised manuscript was clearly explained to us, we could figure out exactly an experimental condition. According to the mussel size and number of mussel tested, the volume of test water and the temperature of the test water, most of the plankton can be assumed to have been fed to the mussels for the two hours of feeding. If possible, it is recommended that chlorophyll fluorescence measure or microscope observation should be checked to see if all the plankton in the test water have been removed before adding the CUR. However, since we were given explicit experimental conditions, we believe this will not be a problem enough to override the test results. Please provide specific numbers in materials and methods for the mean weight of the soft body, gut and gill per mussel (g/individual) used in the analysis so that the author can see fate of the DSTs within the test vessel.

Author Response

Point 1: If possible, it is recommended that chlorophyll fluorescence measure or microscope observation should be checked to see if all the plankton in the test water have been removed before adding the CUR. However, since we were given explicit experimental conditions, we believe this will not be a problem enough to override the test results.

Response: Thank you very much for your professional advice. We shall focus on this issue in our future research.

Point 2: Please provide specific numbers in materials and methods for the mean weight of the soft body, gut and gill per mussel (g/individual) used in the analysis so that the author can see fate of the DSTs within the test vessel.

Response: Thank you very much for your professional reminding. We have supplemented the relevant numbers in materials and methods.

Reviewer 2 Report

The authors have revised the manuscript, correcting the problems and strengthening their conclusions.

Author Response

Thank you very much for your professional and kindly comments.